# Global Transcriptome Analyses of Cellular and Viral mRNAs during HAdV-C5 Infection Highlight New Aspects of Viral mRNA Biogenesis and Cytoplasmic Viral mRNA Accumulations

**DOI:** 10.3390/v14112428

**Published:** 2022-11-01

**Authors:** Margarita Valdés Alemán, Luca D. Bertzbach, Thomas Speiseder, Wing Hang Ip, Ramón A. González, Thomas Dobner

**Affiliations:** 1Department of Viral Transformation, Leibniz Institute of Virology (LIV), 20251 Hamburg, Germany; 2Centro de Investigación en Dinámica Celular, Instituto de Investigación en Ciencias Básicas y Aplicadas, Universidad Autónoma del Estado de Morelos, Cuernavaca 62209, Mexico

**Keywords:** human adenovirus, mRNA biogenesis, mRNA export, next-generation sequencing (NGS), nucleocytoplasmic RNA transport, temporal expression

## Abstract

It is well established that human adenoviruses such as species C, types 2 and 5 (HAdV-C2 and HAdV-C5), induce a nearly complete shutoff of host-cell protein synthesis in the infected cell, simultaneously directing very efficient production of viral proteins. Such preferential expression of viral over cellular genes is thought to be controlled by selective nucleocytoplasmic export and translation of viral mRNA. While detailed knowledge of the regulatory mechanisms responsible for the translation of viral mRNA is available, the viral or cellular mechanisms of mRNA biogenesis are not completely understood. To identify parameters that control the differential export of viral and cellular mRNAs, we performed global transcriptome analyses (RNAseq) and monitored temporal nucleocytoplasmic partitioning of viral and cellular mRNAs during HAdV-C5 infection of A549 cells. Our analyses confirmed previously reported features of the viral mRNA expression program, as a clear shift in viral early to late mRNA accumulation was observed upon transition from the early to the late phase of viral replication. The progression into the late phase of infection, however, did not result in abrogation of cellular mRNA export; rather, viral late mRNAs outnumbered viral early and most cellular mRNAs by several orders of magnitude during the late phase, revealing that viral late mRNAs are not selectively exported but outcompete cellular mRNA biogenesis.

## 1. Introduction

Adenoviruses (AdVs) have long been excellent model viruses not only to study basic virology and cell biology but also for various applications of gene transfer in preclinical and clinical settings [1]. Notably, human AdVs (HAdVs) and primate AdVs are currently being developed or already used as vector vaccines against viral infections such as SARS-CoV-2, HIV, and Ebola virus [2,3,4]. Years of research have led to a thorough understanding of the AdV genome, and the viral replication cycle of the prototypic HAdV-C2 and HAdV-C5 has been studied extensively [1,5]. HAdVs are non-enveloped viruses with a double-stranded linear DNA genome of approximately 36 kilobase pairs (kbp) [6]. Their genome is organized into five early transcription units (TUs) (E1A, E1B, E2, E3, and E4), two intermediate TUs (IX and IVa2), and one late TU (L). Each of these TUs is activated at specific times during the course of viral replication, which progresses through an early and a late phase, separated by the initiation of viral DNA replication. The early and late TUs are all transcribed by the cellular RNA polymerase II; in addition, two short virus-associated (VA) RNAs are transcribed by the RNA polymerase III [6]. Many of the early gene products are multifunctional proteins that regulate gene expression and act as virulence factors. They manipulate a variety of cellular functions, from cell growth and metabolism to the antiviral innate immune response, promoting an optimal DNA replication and gene expression environment, thereby facilitating efficient viral genome replication and progeny production [6,7,8,9]. The viral late genes, L1 to L5, encode structural and other proteins that play regulatory roles in gene expression, and viral progeny assembly [6].

The program of HAdV gene expression follows a complex pattern that has been studied extensively but is still incompletely understood. Transcription of each early gene is controlled by its own promoter, while the late genes are controlled by the major late promoter (MLP) [10,11], and by an internal promoter in the L4 TU independent from the MLP [12]. The activity of each promoter is temporally controlled by a combination of viral and cellular proteins, and both early and late gene products participate in the regulation of every step of viral gene expression, from viral gene transcription (E1A, E1B, E2, E4, IX, IVa2, and L4-22K) [13,14,15,16,17,18,19] to RNA post-transcriptional processing and nucleocytoplasmic export (E1B, E2, E4, L4-22K, and L4-33K) [19,20,21,22,23,24,25,26,27] and mRNA translation (L4-100K, VA RNAs) [28,29,30,31].

A prominent feature of the viral gene expression program is the selective expression of viral genes that results in the almost exclusive synthesis of viral proteins during the course of the late phase of viral replication. Such preferential gene expression is thought to depend on: (i) the inhibition of cap-dependent host cell mRNA translation and efficient translation of viral mRNA and (ii) the inhibition of cellular mRNA nuclear export concurrent with the accumulation of viral late mRNA in the cytoplasm of the infected cell. The mechanism responsible for selective viral mRNA translation is known to depend on the L4-100K protein and the 5’ tripartite leader (TPL) sequence that is common to all HAdV late mRNAs [30,32,33,34,35]. In contrast, the selective cytoplasmic accumulation of viral late mRNAs has yet to be mechanistically explained. Early studies demonstrated that cellular mRNAs fail to accumulate in the cytoplasm of infected cells during the late phase, despite continuous mRNA synthesis and processing [36]. Notably, E1B (E1B-55K) and E4 (E4orf6) gene products are required for the efficient accumulation of newly synthesized viral late mRNAs in the cytoplasm [21,22,37,38,39,40,41,42]. Viral late mRNA transcription and post-transcriptional processing occur in virus-induced nuclear microenvironments known as viral replication compartments (RCs) [27]. Here, E1B and E4 gene products are required for viral late mRNA biogenesis at intranuclear steps that include RC-formation, viral late mRNA splicing, and liberation from RCs, which can be expected to make a major contribution to the efficient export of fully processed viral mRNAs [19,27,43,44,45,46]. Precise mechanisms of viral and cellular mRNA export regulation remain to be described, and a comparative time-dependent global analysis of viral and cellular transcription and nucleocytoplasmic mRNA partitioning has never been performed. To gain new insights into cytoplasmic mRNA accumulation in HAdV-C5-infected cells, we investigated relative abundances of the viral and cellular mRNAs through global high-throughput sequencing analyses from time course infections and nucleocytoplasmic cell fractionation experiments in A549 cells. Our data indicate that (i) abundances of viral early mRNAs increase continuously well after the transition to the late phase of HAdV-C5 replication; (ii) the transition to the late phase of infection is marked by the exponential increase in viral late mRNAs which exceed the levels of all viral early mRNAs by several orders of magnitude; (iii) an unexpectedly large fraction of abundant cellular transcripts can still accumulate in the cytoplasm during the late phase, albeit being outnumbered by viral mRNAs. Our results, therefore, imply that the observed virus-induced cellular mRNA export inhibition during the late phase of HAdV infection is mainly due to extremely efficient viral mRNA expression and export during this phase, rather than a specific and regulated blockage of cellular mRNA export.

## 2. Materials and Methods

### 2.1. Cells and Viruses

Human lung carcinoma A549 cells (ATCC CCL-185; American Type Culture Collection, Manassas, VA, USA) were cultured in Dulbecco’s modified Eagle medium (Sigma, St. Louis, MO, USA) supplemented with 0.11 g/L sodium pyruvate, 10% fetal bovine serum (PAN Biotech, Aidenbach, Germany), and antibiotics (100 U/mL penicillin and 100 μg/mL streptomycin, PAN Biotech) and maintained at 37 °C in a 5% CO_2_ atmosphere. Cells were routinely checked for mycoplasma contamination. For infections, the HAdV-C5 H5*pg*4100 virus was used [47], which has a deletion in the E3 region but a wild-type phenotype in cell culture. We used this virus to ensure comparability to our previous research. Subconfluent A549 cells were infected at a multiplicity of infection of 30 fluorescence-forming units/cell for 6 to 48 h in all experiments. This high multiplicity of infection ensured that all cells were infected and their infection cycles synchronized.

### 2.2. Subcellular Fractionation, RNA Extractions, and High-Throughput RNA Sequencing (RNAseq)

Mock-infected and HAdV-C5-infected cells (~2.5 × 10^6^ per condition) were harvested by trypsinization (at 6, 12, 24, and 48 h p.i.), washed, pelleted, and resuspended in 100 µL cold NP-40 lysis buffer (10 mM HEPES (pH 7.8), 10 mM KCl, 20% (*v*/*v*) glycerol, 0.25% (*v*/*v*) NP-40, and 1 mM DTT) and incubated on ice for 2 min. Samples were centrifuged (470× *g*, 5 min at 4 °C), and the supernatants (~80 μL) were transferred into a new tube containing 600 μL TRIzol (Invitrogen, Carlsbad, CA, USA) and stored as the cytoplasmic fraction at −20 °C. The remaining nuclear pellets were washed with cold NP-40 lysis buffer and centrifuged again, and the supernatants were discarded. These residual pellets, which contained the cell nuclei, were resuspended in 600 μL of TRIzol and stored as the nucleoplasmic fraction at −20 °C until further use. The sum of the mRNA from the nucleoplasmic and cytoplasmic fractions was considered the total pool of mRNA. As a fractionation control, principal component analyses (PCAs) for human reads are provided in Appendix A, where we show that cytoplasmic and nucleoplasmic samples cluster separately. After all of the samples were collected, they were thawed and incubated for 5 min at room temperature; then, 120 µL chloroform was added, and the tubes were shaken for 15 s and incubated for 2–3 min at room temperature. Next, all samples were centrifuged (12,000× *g*, 15 min at 4 °C), and the upper aqueous phase was transferred into a new tube. RNAs were precipitated from these samples by adding 1.5 times the volume of 100% ethanol. From this mixture, 700 µL was transferred to RNA extraction columns (RNeasy Plus Mini Kit (Qiagen, Hilden, Germany)) and further processed according to the manufacturer’s instructions. Final RNA concentrations were measured with a NanoDrop spectrophotometer (Thermo Fisher Scientific, Waltham, MA, USA), and samples were stored at −80 °C. RNA samples from both cytoplasmic and nucleoplasmic fractions were sent to the LIV NGS technology platform. Here, the RNA quality was evaluated using a Bioanalyzer with the RNA Nano Chip (Agilent Technologies, Böblingen, Germany). To isolate mRNAs from the total RNA sample, a poly(A) selection was performed using the NEBnext poly(A) mRNA Magnetic Isolation Module (NEB sequencing, Ipswitch, MA, USA). The cDNA libraries were generated with the ScriptSeq v2 RNAseq Kit (Epicentre Biotechnologies, Madison, WI, USA) using their strand-specific protocol where forward strands were labeled. The sizes and qualities of the libraries were controlled using the Bioanalyzer High Sensitivity DNA Chip Kit (Agilent Technologies). Multiplex sequencing was performed on the HiSeq 2500 platform (Illumina, San Diego, CA, USA) using cDNA library samples at 2 mM and a paired-end run (2 × 100 bp) with a depth of approximately 50 million reads per sample. Quality analyses of the reads were implemented, and adaptor and bar code sequences were removed. Two biological replicates of each infection time point and cellular fraction were sequenced.

### 2.3. Sequence Analyses

The alignment of the sequencing reads to the reference genomes was performed using the CLC Genomics Workbench 9.0 software (Qiagen). All the fastq files were imported to the program in batch mode for each sample, as the multiplex sequencing generated several fastq files per sample. Annotations for the human reference (hg19, GenBank accession number GCA_000001405.1) were downloaded from NCBI to the software, and the HAdV-C5 reference transcriptome was annotated manually (Appendix A). For the alignment, gene and mRNA reference tracks were generated for both reference genomes, and strand specificity was set as “forward” as strands were labeled during the cDNA library preparation. All other settings were set as default. Similarities between sample replicate sequences were assessed by PCAs (Appendix A).

### 2.4. Data Analyses

The table report with read counts and normalized expression values was generated using the CLC Genomics Workbench 9.0 software. Normalized expression values were calculated by the software by dividing the total number of reads per transcript by the length of the transcript, and by the sequencing depth of each sample, resulting in RPKM (reads per kb of transcript per 1 million mapped reads) as the expression unit [48]. For the analyses of the cytoplasmic-to-nucleoplasmic ratios of cellular transcripts, ratios were calculated using RPKM values by applying a Baggerley proportions test to the nucleoplasmic and cytoplasmic expression values of each sample, where group 1 = nucleoplasmic and group 2 = cytoplasmic [49]. All NaN (Not a Number) values were cleared from the data set before analyses, and the Bonferroni- and FDR-corrected *p*-values were calculated for the remaining data. The cytoplasmic-to-nucleoplasmic ratios were used to generate clusters, and the values were represented as a heat map using the R package gplots [50]. Values were z-score transformed and clustered by row (by transcripts, not time points). “Complete agglomeration” was used as the hierarchical clustering method, and “Euclidean distance” was used as the metric. In the case of the analyses of viral transcripts, the comparison between transcripts of each gene and across other viral transcripts was performed by comparing the total reads to avoid the normalization of sequencing depth across infection time points (intrinsic to RPKMs), which would reduce the effect of exponential accumulation of viral transcripts towards the latest time points intrinsic to all viral infections. A comparison of total nuclear and cytoplasmic reads of viral and cellular origin was also performed without normalization. The total expression values were derived by adding the cytoplasmic and nucleoplasmic expression values of each sample.

## 3. Results

### 3.1. Global Analyses of HAdV-C5 Transcripts Reveal Detailed Changes in the Transcription Profiles of Early and Late Transcription Units

We performed a time course infection with HAdV-C5 using A549 cells, which were separated into cytoplasmic and nucleoplasmic fractions for RNA extraction. RNAs were purified and isolated to prepare the cDNA libraries that we analyzed by next-generation sequencing (NGS), as described in Section 2. We first aligned all obtained sequences against the HAdV-C5 transcriptome (Figure 1) to confirm the detection of viral transcripts and to assess whether the selected infection time points reflected a progression in viral replication with distinguishable early and late transcriptional phases. The annotations we used for all viral transcripts were derived by comparing the HAdV-C5 (H5pg4100) genome sequence with the fully annotated HAdV-C2 (AC_000007.1), as they share very high homology [51]. In addition, we included previously reported alternative splice forms of viral mRNAs that are not annotated in the reference sequences [52,53], resulting in 62 annotated HAdV-C5 transcripts (Figure 1A). The experimental setup included infection time points at 6, 12, 24, and 48 h post-infection (h p.i.) (Figure 1B–E). As expected, at 6 and 12 h p.i., most sequences mapped to both ends of the viral genome corresponding to the early TUs E1A, E1B, and E4, and to E2A, E2B, and the truncated E3 TUs (Figure 1B,C), accurately representing the early transcriptional profile of an adenoviral infection. In contrast, at 24 and 48 h p.i., most sequences mapped into the middle section of the genome, corresponding to the major late (ML) TU, including the three small introns that constitute the TPL, which is shared by all viral late mRNAs at the 5′ end [6], indicating that the shift into the late phase has already occurred at 24 h p.i., as expected (Figure 1D,E).

To obtain a global characterization of the HAdV-C5 transcriptome, we performed comprehensive analyses of the steady-state accumulation of viral mRNAs and their relative proportions during HAdV-C5 infection. First, we visualized the overall detection of all HAdV-C5-associated sequences during infection and found an initial accumulation of around 1 × 10^4^ reads at 6 h p.i., followed by an exponential increase in viral sequences in the transition from 12 to 24 h p.i., resulting in an overall abundance of more than 6 × 10^7^ viral reads by 48 h p.i. (Figure 2A). Examination of the origin of those, in relation to their viral TU (Figure 2B), showed that the early units E1A and E4 displayed a constant increase during infection until they plateaued from 24 to 48 p.i.

The E1B and E2 units initially increased until 12 h p.i., after which both displayed a higher exponential increase during the early-to-late phase transition (12 to 24 h p.i.). As expected, the late TU had the lowest contribution of reads during the early phase of infection (6 and 12 h p.i.) and exhibited the largest exponential increase during the transition to the late phase. The accumulation tendencies of each mRNA species reflected the overall proportions of the mRNA species (Figure 2C) when it became evident that transcripts from the early TU were predominant during the early phase until the steep increase in late reads shifted their proportions in the late phase of infection when those were the most abundant transcripts. Interestingly, early transcripts accumulated after the transition to the late phase, until 24 h p.i., and although at lower rates, E1B and E2 continued to increase and represented a noticeable proportion of the total viral mRNA that accumulated in the late phase (Figure 2C).

To examine single viral mRNAs, the expression values were first calculated for the cytoplasmic and nucleoplasmic fractions, and the total mRNA expression values were defined to be the sum of both, at each time point of infection. We observed that most individual viral transcripts showed steady-state accumulations with similar tendencies as the other mRNAs in the same TU (Figure 2D,F,H,J,L). However, a few noticeable transcripts had a different expression pattern, which affected the overall proportion of their relative abundance in relation to their TU (Figure 2E,G,I,K,M), namely E1A 9S, E1B 22S, E2A DBP, and E2B IVa2 and the E4orf6/7 late mRNA, which displayed either higher or lower levels of accumulation through the course of viral replication. For early E1A transcripts, the E1A 12S and 13S mRNAs were the most abundant transcripts, but a clear shift was observed after the transition to the late phase when the E1A 9S increased exponentially and was present as abundantly as E1A 12S and 13S (Figure 2D,E). The E1B TU was almost completely represented by the E1B 22S (55K) during the early phase (> 95%), with already approximately 4 × 10^4^ reads at 6 h p.i. The exponential increase in the smaller E1B isoforms in the late phase, in particular the E1B 13S and 14.5 S, resulted in a higher representation of these transcripts (~25%) during the late phase (Figure 2F,G). Noteworthy was the shift within the E2 TU in the early-to-late transition. While the E2A DBP transcript was the most abundant in the early phase (from ~70 to 77%), this mRNA rapidly decreased after 12 h p.i., concurrent with an exponential increase in the E2B IVa2 transcript, which represented close to 75% of all E2 transcripts at 24 h p.i. (Figure 2H,I). Transcripts from the E4 TU, although expressed at different levels, displayed similar non-exponential accumulation tendencies. Nevertheless, the E4orf6/7_2 late mRNA showed a steeper increase between 12 and 24 h p.i., transitioning from only ~5% in the early phase to ~25% of all E4 transcripts by 24 h p.i. (Figure 2J,K). The late TU clearly exhibited the largest overall increase in the early-to-late transition and represented the highest number of viral reads during the late phase (~85–95%). All late mRNAs showed an exponential accumulation, with the L5 pIV fiber and L3 hexon mRNAs reaching the highest levels during the transition to the late phase and continuing to increase, accounting for 10^7^ reads at 48 h p.i. (Figure 2L,M).

In summary, these data confirmed the expected changes in the levels of viral mRNAs during the transition from the early phase to the late phase of infection. More specifically, we observed an increase in viral late mRNAs and an apparent downregulation of early mRNAs, considering their overall proportions (Figure 2C). However, even though early transcripts were 10^2^ to 10^3^ times less abundant than the viral late transcripts (Figure 2B), the global and detailed analyses of the temporal changes in individual early mRNA species, which may have been overlooked so far due to the overwhelming increase in viral late mRNAs, revealed that most early mRNA species continued to accumulate during the late phase.

### 3.2. Analysis of Viral and Cellular mRNA Cytoplasmic-to-Nucleoplasmic Ratios Shows No Global Decrease in the Cytoplasmic Accumulation of Cellular Transcripts during the Late Phase of Infection

To study the effects of HAdV-C5 infection on cytoplasmic accumulations of cellular mRNAs, we performed a global examination of the cytoplasmic to nucleoplasmic cellular mRNA ratios (cyto/nuc) focusing on the transition from the early to late phase of infection (see Section 2), considering that our earlier analyses indicated that this transition period appears to be at 12–24 h post-infection. The data were selected by filtering with a false discovery rate (FDR)-corrected *p*-value of ≤0.05, obtaining a total of 737 cellular transcripts. To focus the analysis on cyto/nuc ratios with significant changes during infection, ratios with greater than 2-fold changes between at least two sampling time points p.i. were selected. This adjusted dataset included 339 cellular mRNAs for which the nucleocytoplasmic partitioning was significantly altered during infection. The cellular mRNAs with similar changes in cyto/nuc ratios were clustered, and we generated a heat map to visualize changes at different time points p.i. (Figure 3). The clustering analysis revealed three main branches (I–III) depicting the main cyto/nuc variation patterns at different time points p.i. From these main branches, we identified seven distinct clusters (A–G) (Appendix A). Overall, the data showed considerable changes in the cyto/nuc ratios of cellular mRNAs during infection. First, most mRNA cyto/nuc ratios were inverted when comparing 6 to 24 h p.i., with the 12 h p.i. time point appearing to be a transition phase. Second, cyto/nuc ratios of mRNAs nicely clustered together, suggesting that the mRNAs of certain clusters accumulate more efficiently in the cytoplasm during the early phase of infection (Figure 3, clusters A, B, and C), while a large set of cellular mRNAs had higher cyto/nuc ratios at 24 h p.i. (Figure 3, clusters F and G), indicating that the latter mRNAs accumulate in the cytoplasm more efficiently during the late phase of infection. Hence, cellular mRNA export is not blocked during the late phase of viral replication. The data rather show that the transition from the early to the late phase results in extensive variations in the cellular mRNA species that are exported to the cytoplasm.

### 3.3. Comparison of the Distribution of Viral Late and Cellular mRNA Reveals Markedly Increased Levels of Cytoplasmic Viral Late RNAs

Since we did not observe a complete block in the cytoplasmic accumulation of cellular mRNAs during the late phase of infection, we decided to use our methodological setup to gain insight into what has previously been reported as a blockage of cellular mRNA export, which contributes to the viral shutoff of host protein synthesis in HAdV infection [54]. We, therefore, further analyzed the global mRNA dynamics during infection by comparing the relation between nuclear and cytoplasmic transcript levels, and their changes during the course of infection, for both viral and cellular reads. 

As demonstrated previously, viral transcripts increased exponentially in the transition to the late phase (Figure 2A). In the case of cellular transcript levels, they were found to be relatively constant, at roughly 1.4 × 10^7^ to 2.5 × 10^7^ reads throughout HAdV-C5 replication (Figure 4B). As described above, the number of viral transcripts was found to be around 1 × 10^4^ to 1 × 10^5^ reads in the early phase (Figure 2A). With their exponential increase in the late phase, they reached comparable levels to cellular mRNAs and eventually even exceeded them (~1.4 × 10^7^ to ~5 × 10^7^ reads) (Figure 4A). Furthermore, the proportion between cytoplasmic and nucleoplasmic reads at each time point of infection showed that prior to the transition to the late phase, more than half of the total reads were cellular cytoplasmic reads (~53–61%), with a very small proportion of viral cytoplasmic reads (~0.1%), which are not visibly noticeable in our graphs (Figure 4C, 6 and 12 h p.i.). A noteworthy shift in cellular and viral cytoplasmic proportions was observed during the late phase, as a consequence of the exponential increase in viral reads. By 24 h p.i., viral cytoplasmic were equivalent to cellular cytoplasmic reads, and by 48 h p.i., they were 6 times more abundant than all cellular reads (Figure 4C, 24 and 48 h p.i.). Changes in the global cytoplasmic-to-nucleoplasmic proportions were also observed during the late phase, with only a small decrease in the percentage of the cytoplasmic cellular reads, from 46.5% to 42.7% (Figure 4B). In contrast, viral transcripts in the cytoplasm increased from 56.8% to 72% between 24 and 48 h p.i. (Figure 4A). In summary, these data indicate that almost all cytoplasmic mRNAs were of cellular origin during the early phase of infection, whereas the proportions of total cellular transcripts decreased as total viral transcripts increased exponentially from 12 to 24 h p.i. (Figure 4A). Interestingly, nearly 86% of the total cytoplasmic mRNAs were virus-specific at 48 h p.i. (Figure 4C, 48 h p.i.), but the proportions between cellular nuclear and cytoplasmic fractions were not significantly altered.

Overall, our data show that only some cellular cyto/nuc transcript ratios are downregulated during the late phase, while others are upregulated (Figure 3); however, although up to 2-fold variations in the levels of cellular mRNA between mock-infected cells and the different time points p.i. were observed, we did not discern a significant effect on the total cellular cytoplasmic vs. nucleoplasmic ratios. During infection, the total cellular mRNAs were maintained at a scale of 10^7^ reads, while total viral mRNAs increased, reaching cellular levels late in infection. As expected, HAdV infection did not induce a global effect on the level of total cellular mRNA (Figure 4B), and cellular mRNA export was not blocked. Thus, the higher efficiency of viral mRNA accumulation (Figure 4C) in the cytoplasm may be due to more efficient intranuclear processing steps [43]. Nevertheless, since high levels of cellular mRNAs were still produced even at late time points p.i., we decided to compare the viral late mRNAs and cellular mRNAs in terms of relative abundance throughout infection. Therefore, the top 35 most abundant cellular mRNAs were compared with all viral late mRNAs at each time point p.i. (Figure 5). Interestingly, although individual mRNA level fluctuations were apparent, the numbers of reads corresponding to the most abundant cellular mRNAs were between ~5 × 10^4^ and 5 × 10^6^ throughout infection, while viral late mRNAs reached similar numbers at 24 h p.i., reaching levels above 1 × 10^6^ and 1 × 10^7^ reads by 48 h p.i. (Figure 5, 24–48 h p.i.). These data show that viral late mRNAs can reach levels exceeding most cellular mRNAs during the late phase, reaching comparable levels with the most expressed cellular mRNAs. Moreover, we found that cellular mRNAs can still be produced and exported at comparable levels (Figure 4B) while viral late mRNAs are produced more efficiently at late time points p.i. (Figure 4A). Taken together, our findings revealed that cellular mRNA export is not blocked during the late phase of infection and suggest that the higher efficiency of viral late mRNA accumulation in the cytoplasm (Figure 4C) results from the higher accessibility of viral mRNA to post-transcriptional processing factors that are recruited to the viral RCs where the viral mRNAs are produced and processed in the nucleus, making their coupling to the mRNA export pathway highly efficient [55,56,57,58,59,60].

## 4. Discussion

Understanding the dynamic changes in viral mRNA expression during HAdV infection has long been of interest to adenoviral research. Several experimental approaches aimed at deciphering changes in transcriptional profiles during HAdV infection have found that viral late mRNAs are favorably transported from the nucleus to the cytoplasm while cellular mRNAs are excluded from this process. Previous work either only focused on a few distinct RNA species [61,62,63] or did not quantitatively compare the contribution of each viral or cellular mRNA to the total pool of transcripts, between the nucleus and cytoplasm, at different stages of the viral replication cycle due at least in part to the lack of high-throughput analysis methods in the 1970s and 1980s [21,22,36,38,43,64]. Thus, to date, nuclear export of only a small number of cellular mRNAs (e.g., Hsp70, ß-actin, tubulin, 6–16, and Mx-A) has been studied in comparison to nuclear export of highly abundant viral late mRNAs during HAdV infection. These studies used different cells and, in part, different adenoviruses as well as mutants, which reduces comparability to our data. However, we found actin and certain heat shock protein mRNAs in our top 35 most abundant cellular transcripts at different time points p.i. (Appendix A), contradicting previous data that reported lower levels of these mRNAs during later phases of viral replication. Very recent HAdV RNAseq data provided insights into splicing events and alternative splice isoforms in the complex HAdV transcriptome by long-read RNAseq and mapping of METTL3-mediated m6A modifications [65,66,67,68]. However, no transcriptome analyses of HAdV infection comparing viral and cellular RNAs and/or cytoplasmic and nucleoplasmic fractions during infection have been reported. In this work, we studied the transcriptional profile of both viral and cellular mRNAs and examined the effects of HAdV-C5 infection on mRNA biogenesis and cytoplasmic mRNA accumulation at different time points p.i. by NGS.

We identified 62 HAdV-C5 transcripts and determined their accumulation during infection. In addition, we determined the cytoplasmic and nucleoplasmic proportions of cellular transcripts at different time points p.i. By analyzing time courses of viral mRNA abundances during HAdV infection, we showed that early during infection, all reads mapped to the ends of the viral genome and the E2 TU (Figure 1). High levels of E1 to E4 transcripts in the early phase of infection were also found in previous reports using different approaches such as R-loop mapping and RNA–DNA hybridization [69,70]. The late phase of infection is then characterized by the tremendously increasing number of reads that align to the ML TU and to the TPL, again in agreement with previous work [68,69]. The increase in total viral reads from 6 to 12 h p.i. (Figure 1 and Figure 2) is very likely an effect of the transcriptional activation by E1A [71,72,73].

Overall, most viral early mRNA expression patterns were comparable between mRNAs within the same TU, and while most E1A and the E4 mRNAs gradually increased during infection, E1B and E2B IVa2 mRNAs exponentially increased by more than two orders of magnitude (Figure 2). Notably, a pair of E2 mRNAs, E2B IVa2 and E2A DBP, displayed a unique expression pattern (Figure 2E) that showed opposing expression regulation in the transition to the late phase of infection, where DBP mRNA was downregulated whereas E2B IVa2 was exponentially upregulated (Figure 2E). DBP expression is essential for various steps of HAdV replication and progeny production [14,20,74,75,76,77,78]. Similar to our observations, E2A region transcripts have also been found to be low in other reports [70,79]. The decline in E2A DBP mRNA does not seem to match the increase in viral DNA synthesis. However, DBP protein levels have been documented extensively and are known to increase steadily until late times p.i. Therefore, further experiments are required to evaluate the E2A DBP mRNA and protein expression patterns as the viral replication cycle progresses, as well as the mechanism that may be responsible for the mRNA decline after the initiation of viral DNA replication. A decrease in DBP transcripts during infection could be compensated by the observation that the protein itself seems very stable [80,81]. Moreover, tight control of mRNA stability is required for adenoviral gene regulation and could be an explanation for these observations [82]. Lutz et al. demonstrated that E2B IVa2 is required for activation of late gene expression and its transcription marks the transition from the early to the late phase of HAdV infection, a likely explanation for the observed increase in E2B IVa2 mRNAs in the transition to the late phase of infection in our analyses [17,18]. Therefore, the switch between E2A DBP and E2B IVa2 mRNAs could be important for the transition into the late phase of infection. Previously, viral early mRNA synthesis was thought to decline or even stop during the late phase of infection, but the steady-state levels of early mRNAs detected here and in previously reported data [83] demonstrate that most early transcripts do not decrease as the infection progresses into the late phase. On the contrary, we not only prove that most viral early mRNA levels steadily increase during the late phase of infection (at least up to 24 h p.i.), but also show that due to their pattern of accumulation there is a switch in their expression compared to the early phase.

Transcription of viral late mRNAs was scarce during the early phase of infection, accounting for roughly 5% and 9% of all detected viral reads at 6 and 12 h p.i., respectively, which is in agreement with previous findings [36]. Between 12 and 24 h p.i., viral late mRNA expression increased exponentially, resulting in levels that exceeded all early mRNAs by several orders of magnitude as the infection progresses, to the point where 94% of all viral mRNAs correspond to the ML TU (Figure 2 and Figure 4). The comparison between cellular and viral transcripts during infection revealed that only 0.07% and 0.3% were viral at 6 and 12 h p.i., respectively. These proportions dramatically changed during the transition to the late phase of infection when viral mRNA levels exponentially increased, reaching levels of roughly 43% and 79% of all transcripts at 24 and 48 h p.i. (Figure 5). A remarkable increase in viral over cellular mRNAs has also been described in very early HAdV research [84,85,86]. Analyses of the cytoplasmic accumulation of host cell transcripts during infection revealed that their abundances did not change significantly between different time points. Interestingly, the early observations that cellular mRNAs are restricted to the nucleus showed that, depending on the methods used, between 80 and up to 98% of the newly synthesized poly(A)-containing RNAs in the cytoplasm during the late phase were viral-specific [36,84,85]; therefore, it is possible that the high sensitivity in our experiments affords a higher level of precision and accounts for the differences observed. We found sets of host cell mRNAs that either decreased or increased in cytoplasmic accumulation in the transition to the late phase of infection (Figure 3). These observations argue against a global shutoff of cellular mRNA export during the late phase of infection, and rather indicate that the viral mRNAs are produced and processed more efficiently than the cellular mRNAs. Interestingly, when searching for functional associations to the transcripts in the up- and downregulated clusters, both showed terms related to RNA splicing, RNA binding, and protein synthesis, among others (data not shown). Thus, in contrast to what has been previously interpreted, we discern that viral late mRNAs and a number of cellular mRNAs accumulate in the cytoplasm at late time points p.i. We also hypothesize that the predominance of viral mRNAs at late stages of infection does not result from an export shutoff of cellular mRNAs, but rather from more efficient intranuclear post-transcriptional processing associated with their site of synthesis and splicing in the viral RCs [55,56,57,58], in agreement with previous findings [27,43]. The absence of cellular mRNA export shutoff notwithstanding, the extensive changes in the cyto/nuc ratios of cellular mRNA at different time points suggest that adenovirus infection considerably alters the patterns of cellular gene expression at the transcriptional and/or post-transcriptional levels and should be explored further.

The E1B-55K/E4orf6 complex has been previously described to be required for an efficient transition into the late phase and accumulation of viral late mRNAs in the cytoplasm [87]. Although previous data suggested that this complex could act as a nucleocytoplasmic shuttle for viral mRNAs, no solid experimental proof has been provided yet for a direct role of these proteins during the export of mRNA. Abrogation of CRM-1-dependent export of either E1B-55K or E4orf6 confirmed previous findings showing that the proteins’ nucleocytoplasmic export is not necessary for viral late gene expression [45,63,88,89,90], but nucleocytoplasmic export and degradation of key cellular targets of the E1B-55K/E4orf6 dependent E3 ubiquitin ligase was affected by mutations of the proteins’ NES [45]. E4orf6 has been implicated in the regulation of alternative splicing of the viral late mRNA tripartite leader [86], and viral late mRNA splicing was recently shown to be affected by E1B-55K RNA binding [91]. Efforts to investigate its contribution to HAdV mRNA biogenesis and intracellular transport using the latest sequencing technologies have recently revealed that E1B-55K regulates viral gene expression, as well as the release of viral late mRNAs from their site of synthesis (viral RCs) in a phosphorylation-dependent fashion [19]. Therefore, the role of these important viral proteins in cellular protein degradation, RC formation, and regulation of intranuclear steps of viral gene expression should be studied in further detail.

In summary, our data reveal novel aspects of adenoviral mRNA biogenesis and highlight higher efficiency of intranuclear post-transcriptional processing rather than shutoff of cellular mRNA export as the predominant mode of cytoplasmic mRNA accumulation in HAdV-C5 infection.

## Figures and Tables

**Figure 1 viruses-14-02428-f001:**
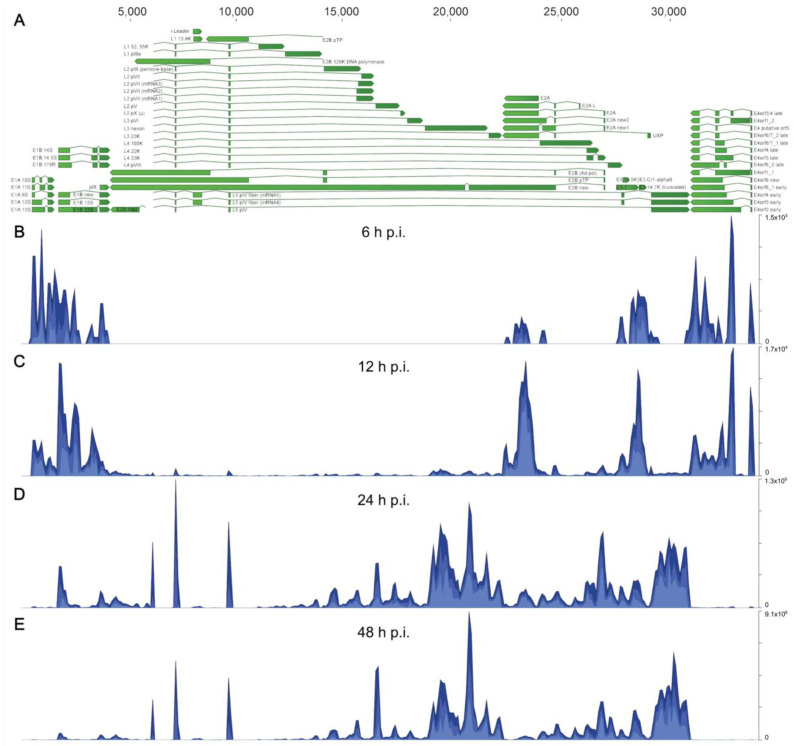
The HAdV-C5 transcriptome in A549-infected cells at different time points post-infection. Visualization of total (nuclear and cytoplasmic) mapped sequences across the HAdV-C5 transcriptome at each time point p.i. (**A**) The annotated transcriptome of HAdV-C5 is shown, represented as green arrows, with each annotated transcript belonging to each of the adenoviral gene families (E1A, E1B, E4, E2A, E2B, and ML (L1–L5)), including the remnants of the E3 region, which is partially deleted in the H5pg4100 virus that was used as “wild-type” HAdV-C5 (see Section 2.1). The directions of arrows represent the coding sequence direction for each viral transcript. Nucleotide position annotations for all transcripts are listed in Appendix A. (**B**–**E**) Histograms show cumulative coverage of sequencing reads that were mapped to the viral genome at (**B**) 6, (**C**) 12, (**D**) 24, and (**E**) 48 h p.i. The scale, in total reads, for every histogram is shown on the right.

**Figure 2 viruses-14-02428-f002:**
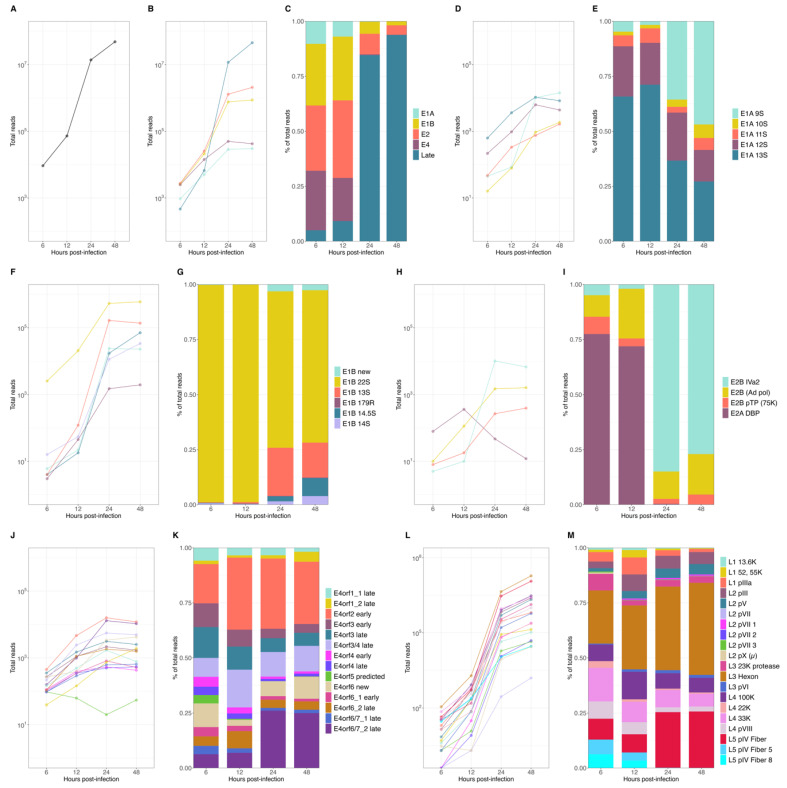
Expression levels of HAdV-C5 mRNAs at early and late time points p.i. Different time points p.i. of (**A**) total viral mRNA expression values, (**B**) changes in abundance of the different viral mRNA species produced from each TU, and (**C**) corresponding proportions of all viral reads. Analyses of changes in abundance of mRNAs and the corresponding relative proportions were performed for individual transcripts of each viral transcription unit: E1A (**D**,**E**), E1B (**F**,**G**), E2 (**H**,**I**), E4 (**J**,**K**), and ML (**L**,**M**).

**Figure 3 viruses-14-02428-f003:**
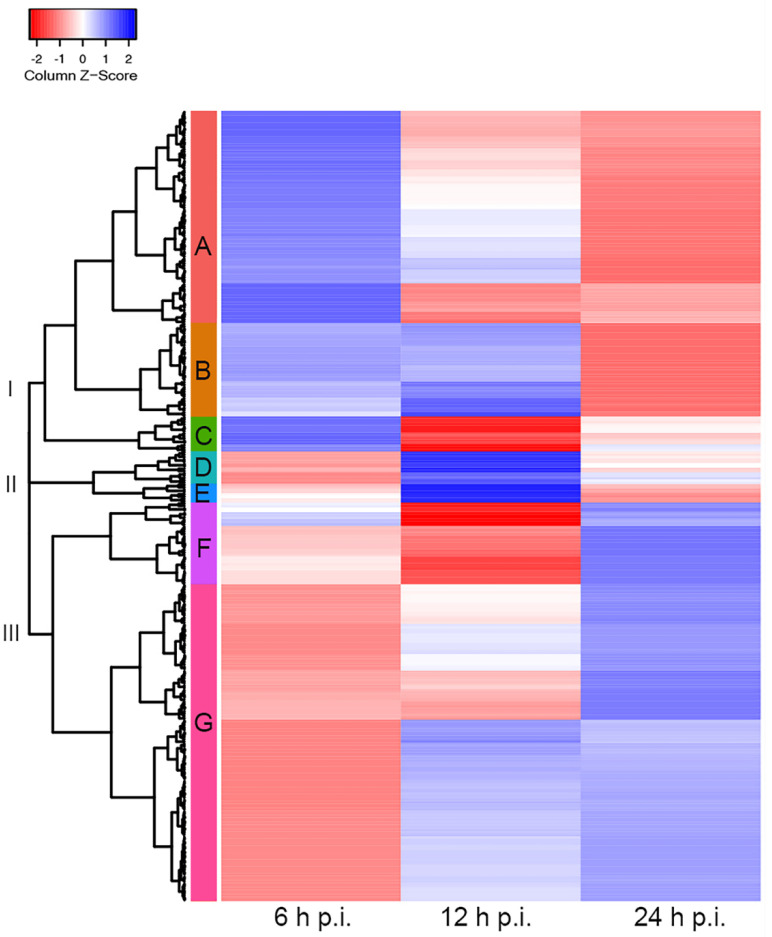
Cellular cytoplasmic/nucleoplasmic mRNA ratios at different time points post-infection. Cluster heat map of cellular transcripts with FDR-corrected *p*-values (≤0.05) and fold changes of ≥2. Cytoplasmic-to-nucleoplasmic ratios of cellular mRNAs were z-score transformed and clustered applying the Euclidean distance method. The resulting tree includes three main branches (I–III) and seven clusters (A–G). Blue colors indicate mRNAs with higher cytoplasmic abundance, and mRNAs that are more abundant in the nucleoplasm are represented in the red z-score spectrum. Transcripts associated with each cluster are listed in Appendix A.

**Figure 4 viruses-14-02428-f004:**
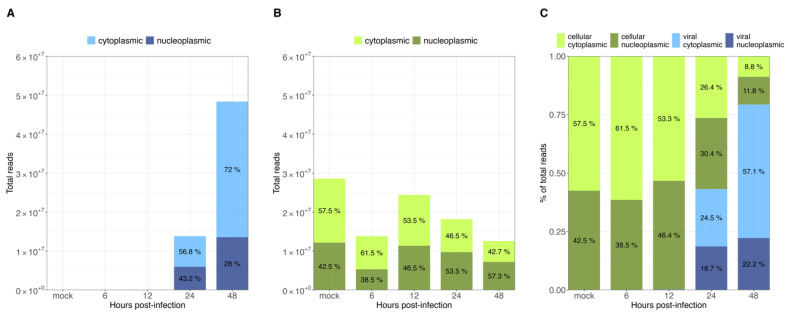
Intracellular distribution of viral and cellular mRNAs at different time points post-infection. All RNAseq reads were assigned to (**A**) viral and (**B**) cellular transcripts. Total reads at each time point p.i. are represented as bar graphs. (**C**) Proportions of cytoplasmic or nucleoplasmic reads are shown for each time point p.i.

**Figure 5 viruses-14-02428-f005:**
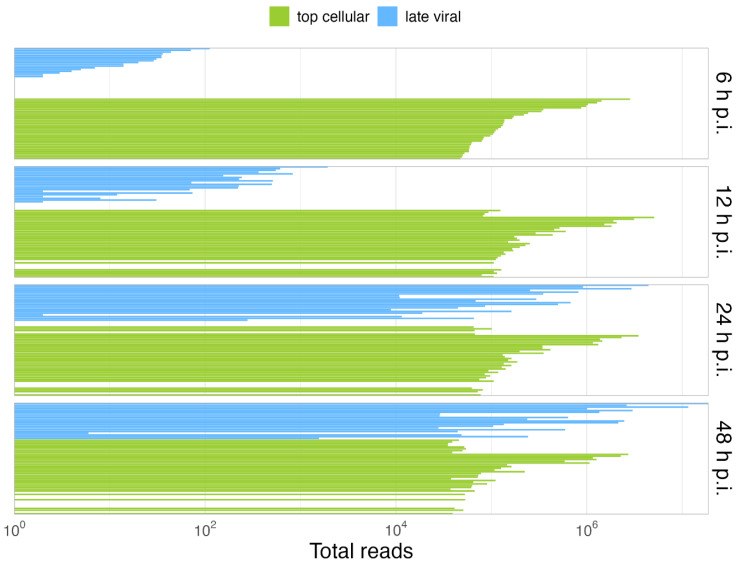
Viral late mRNAs are comparable in abundance to the most abundant cellular mRNAs in the late phase. Shown is the comparison of the relative abundances of the 35 most abundant cellular mRNAs (green) vs. the viral late mRNAs (blue), at each time point p.i. A detailed list of the cellular top transcripts and their rank in terms of abundance can be found in Appendix A.

## Data Availability

Please contact the corresponding author if you wish to request the RNAseq raw data generated for this study.

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
