# Peer review of "Global Transcriptome Analyses of Cellular and Viral mRNAs during HAdV-C5 Infection Highlight New Aspects of Viral mRNA Biogenesis and Cytoplasmic Viral mRNA Accumulations"

_viruses, 2022, doi:10.3390/v14112428_

Round 1

Reviewer 1 Report

Viral mRNA translation is more efficient than that of host messengers in species C human adenovirus (HAdV-C)-infected tissue culture cells at late phase of virus infection. It has been assumed that this is at least partially due to retention of host mRNAs in the nucleus in the late phase of infection. However, this  hypothesis was build on analyses of only few host mRNAs. Valdés Alemán et al. have now taken a more global view of nucleus vs cytoplasmic distribution of viral and host mRNAs at different time points post infection in HAdV-C5-infected A549 cells using RNA extraction from cytoplasmic and nucleoplasmic cellular fractions and next-generation sequencing from cDNA libraries. Their bulk RNA-seq confirms the overall kinetic early-late distribution of viral transcripts, and comparison of total virus vs cellular transcript reads at different times pi nicely demonstrated that whereas early viral transcripts constituted <1% of all transcript reads at 6 and 12 h pi, there was a dramatic increase in the number of viral transcripts once the infection had proceeded into the late phase, 79% of all transcripts being of viral origin at 48 h pi. Interestingly, although some cellular transcripts displayed increasing nuclear retention at late phase of infection, this was not a general trend since a large set of cellular mRNAs accumulated even more efficiently in the cytoplasm at late infection phase than at the early phase. The Authors conclude that there is no global shutoff of cellular mRNA export during the late phase of virus replication. This is an important finding. However, a major criticism to the study is that the Authors do not offer any controls that their subcellular fractionation yields pure nucleoplasmic and cytoplasmic fractions. Their conclusions are critically dependent on these controls. The Authors could look at their RNAseq data and see whether intron reads of cellular transcripts (if present in sufficient numbers) are absent from their cytoplasmic fraction, and/or the Authors could do Western blot analyses from their cytoplasmic and nucleoplasmic fractions (prior to TRIzol extraction) for select nuclear and cytoplasmic proteins.

Minor comment:

p.12, starting from line 459: “We also conclude that the predominance of viral mRNAs at late stages of infection does not result from an export shutoff of cellular mRNAs, but rather from more efficient intranuclear posttranscriptional processing associated with their site of synthesis and splicing in the viral RCs [55-58],….”. The first part is indeed true if their subcellular fractionation yields clean nucleoplasmic and cytoplasmic fractions, but there is no direct evidence for the latter part, ie that viral transcripts would have more efficient intranuclear posttranscriptional processing than host transcripts. Therefore, it would be better to tune this latter part down a bit, eg. to use the verb “suggest” or “hypothesize” instead of “conclude” for this part.

Author Response

Comment 1: Viral mRNA translation is more efficient than that of host messengers in species C human adenovirus (HAdV-C)-infected tissue culture cells at late phase of virus infection. It has been assumed that this is at least partially due to retention of host mRNAs in the nucleus in the late phase of infection. However, this hypothesis was build on analyses of only few host mRNAs. Valdés Alemán et al. have now taken a more global view of nucleus vs cytoplasmic distribution of viral and host mRNAs at different time points post infection in HAdV-C5-infected A549 cells using RNA extraction from cytoplasmic and nucleoplasmic cellular fractions and next-generation sequencing from cDNA libraries. Their bulk RNA-seq confirms the overall kinetic early-late distribution of viral transcripts, and comparison of total virus vs cellular transcript reads at different times pi nicely demonstrated that whereas early viral transcripts constituted <1% of all transcript reads at 6 and 12 h pi, there was a dramatic increase in the number of viral transcripts once the infection had proceeded into the late phase, 79% of all transcripts being of viral origin at 48 h pi. Interestingly, although some cellular transcripts displayed increasing nuclear retention at late phase of infection, this was not a general trend since a large set of cellular mRNAs accumulated even more efficiently in the cytoplasm at late infection phase than at the early phase. The authors conclude that there is no global shutoff of cellular mRNA export during the late phase of virus replication. This is an important finding. However, a major criticism to the study is that the authors do not offer any controls that their subcellular fractionation yields pure nucleoplasmic and cytoplasmic fractions. Their conclusions are critically dependent on these controls. The authors could look at their RNAseq data and see whether intron reads of cellular transcripts (if present in sufficient numbers) are absent from their cytoplasmic fraction, and/or the authors could do western blot analyses from their cytoplasmic and nucleoplasmic fractions (prior to TRIzol extraction) for select nuclear and cytoplasmic proteins.

Response 1: We cordially thank the reviewer for their careful evaluation of our work. To address their major criticism regarding the question if our subcellular fractionation protocol yields pure nucleoplasmic and cytoplasmic fractions, we would kindly like to point towards our Supplementary Figure 1, where we show principal component analyses (PCAs) for both the human and viral reads that we detected. PCAs allow transformation of high-dimensional data, such as high-throughput sequencing data, into low-dimensional data. In these graphs, each dot represents all the reads from a sequenced sample condition. Here, samples that have similar expression profiles nicely cluster together. When assessing human reads, we can observe two large clusters, one with all nuclear samples and the other with all cytoplasmic samples, separated along the x-axis, representing the greatest variance (36%) in our analyses. Therefore, we can infer that our cytoplasmic and nucleoplasmic samples are indeed different subcellular fractions, as expected. We would like to note that PCAs are standard controls for sequencing analyses and allow an assessment of differences or similarities among sequenced samples i.e. different cell types, conditions, controls, replicates, etc.

We now included an explanation in lines 122f. This text was added: “As a fractionation control, principal component analyses (PCAs) for human reads are provided in Fig. S1, were we show that cytoplasmic and nucleoplasmic samples cluster separately.”

Comment 2: p.12, starting from line 459: “We also conclude that the predominance of viral mRNAs at late stages of infection does not result from an export shutoff of cellular mRNAs, but rather from more efficient intranuclear posttranscriptional processing associated with their site of synthesis and splicing in the viral RCs [55-58],….”. The first part is indeed true if their subcellular fractionation yields clean nucleoplasmic and cytoplasmic fractions, but there is no direct evidence for the latter part, ie that viral transcripts would have more efficient intranuclear posttranscriptional processing than host transcripts. Therefore, it would be better to tune this latter part down a bit, eg. to use the verb “suggest” or “hypothesize” instead of “conclude” for this part.

Response 2: We completely agree with the reviewer and changed the sentence accordingly.

Reviewer 2 Report

In their manuscript titled “Global Transcriptome Analyses of Cellular and Viral mRNAs During HAdV-C5 Infection Highlight New Aspects of Viral mRNA Biogenesis and Cytoplasmic Viral mRNA Accumulations”, Valdes Aleman et al. describe changes in viral and cellular mRNA abundance during the course of adenovirus infection.  The authors find that the overall decrease in the proportion of host cytoplasmic mRNA level compared to the viral one is due to the increased accumulation of viral messages and not due to the global inhibition of host mRNA export as it was reported previously.  They explain this discrepancy with their use of a technology that allowed them to obtain a more comprehensive view of the changes.  

Critique

1.       The cytoplasmic abundance of some messages has been shown to decrease at the late phase of infection (they list several; lines 389 to 392).  What are changes they found for those messages?  Are they consistently down in the cytoplasm?  That is, are the fundamental findings of this manuscript consistent with those of multiple previous publications, and only new data not accessible to previous authors led to a novel conclusion?  These comparisons to older data need to be shown and discussed.

2.       An E3-deleted mutant was chosen for the infection.  What was the rationale for this choice?  Although the E3 proteins have not been implicated in mRNA transport thus far, it cannot be excluded that the novel technique used could have found a link.  Further, it would have been interesting to see the expression pattern of the various E3 messages.  The reasons of this choice and its implications need to be made clear in the paper.  Calling the deletion mutant “wild type HAdV-C5” is not acceptable and should be corrected.

3.       For the results described in Fig. 1, it should be clarified whether they represent total (nuclear and cytoplasmic) mRNA.

4.       Omitting the scale from Fig. 1 is misleading; the reader can conclude that most early messages almost completely disappear by 24 h post infection.  While data shown in later figures clearly show that this is not the case, showing the scale and pointing out differences in it in the figure legend would clarify this.

5.       Lines 435 to 437:  Make it clear that the late messages are 5 and 9 percent of all viral reads, not all reads.

Author Response

Comment 1: In their manuscript titled “Global Transcriptome Analyses of Cellular and Viral mRNAs During HAdV-C5 Infection Highlight New Aspects of Viral mRNA Biogenesis and Cytoplasmic Viral mRNA Accumulations”, Valdes Aleman et al. describe changes in viral and cellular mRNA abundance during the course of adenovirus infection.  The authors find that the overall decrease in the proportion of host cytoplasmic mRNA level compared to the viral one is due to the increased accumulation of viral messages and not due to the global inhibition of host mRNA export as it was reported previously.  They explain this discrepancy with their use of a technology that allowed them to obtain a more comprehensive view of the changes. 

Response 1: We thank the reviewer for their comments, which we believe accurately describe this paper’s contribution.

Comment 2: The cytoplasmic abundance of some messages has been shown to decrease at the late phase of infection (they list several; lines 389 to 392). What are changes they found for those messages?  Are they consistently down in the cytoplasm?  That is, are the fundamental findings of this manuscript consistent with those of multiple previous publications, and only new data not accessible to previous authors led to a novel conclusion?  These comparisons to older data need to be shown and discussed.

Response 2: Thank you for this comment, we now elaborate on that in lines 401f.

We did not find significant changes for the previously analyzed cellular mRNAs, the only significant changes in mRNA ratios are the ones we show in Figure 3 and Tab S2. However, it is difficult to make accurate comparisons with previous data since they used different techniques, different cell-types, and especially compared cellular mRNA against viral late mRNAs across different mutant viruses, which was different from our approach.

This text was added: “These studies used different cells and, in part, different adenoviruses as well as mutants, which reduces comparability to our data. But we found actin and certain heat shock protein mRNAs in our top 35 most abundant cellular transcripts at different time points p.i. (Tab. S3), contradicting previous data that reported lower levels of these mRNAs during later phases of viral replication.”

Comment 3: An E3-deleted mutant was chosen for the infection.  What was the rationale for this choice?  Although the E3 proteins have not been implicated in mRNA transport thus far, it cannot be excluded that the novel technique used could have found a link.  Further, it would have been interesting to see the expression pattern of the various E3 messages.  The reasons of this choice and its implications need to be made clear in the paper.  Calling the deletion mutant “wild type HAdV-C5” is not acceptable and should be corrected.

Response 3: This is a good comment. The H5pg4100 virus was developed by P. Groitl and T. Dobner in 2007 and has been used in several research groups studying adenovirus, it is equivalent to the dl309 adenovirus, developed by N. Jones and T. Shenk, which has been used since 1979 for adenovirus studies. Both have a deletion in the E3 region but show a wild-type phenotype in cell culture assays. We changed the text accordingly to briefly explain our rationale for using this virus in line 104f.

Comment 4: For the results described in Fig. 1, it should be clarified whether they represent total (nuclear and cytoplasmic) mRNA.

Response 4: Done, thanks!

Comment 5: Omitting the scale from Fig. 1 is misleading; the reader can conclude that most early messages almost completely disappear by 24 h post infection.  While data shown in later figures clearly show that this is not the case, showing the scale and pointing out differences in it in the figure legend would clarify this.

Response 5: We agree with the reviewer and changed Fig. 1 and its legend accordingly. The y-axis showing the scale for the histograms, in total reads for each time point p.i. are now included in the figure and mentioned in the legend.

Comment 6: Lines 435 to 437: Make it clear that the late messages are 5 and 9 percent of all viral reads, not all reads.

Response 6: Corrected.

Round 2

Reviewer 1 Report

The Authors have adequately answered the concerns raised in the primary round of review and the revised manuscript is ready for publication.

Author Response

Thanks, perfect!

Reviewer 2 Report

1. As the data in the manuscript directly contradicts previously reported results, the authors should validate their findings using another method besides RNASeq.  They should use RTqPCR to quantify cytoplasmic and nuclear messages for actin, the export of which was reported to be suppressed. 

2. For Fig. 1, please, add a Y axis and an actual scale to the graph; showing just 0 and the top value is hard to follow.  Also, delete the "wild type" designation for  H5pg4100 in the figure legend; a deletion mutant of a virus is not wild type.
